# Modulation of Adipocyte Metabolism by Microbial Short-Chain Fatty Acids

**DOI:** 10.3390/nu13103666

**Published:** 2021-10-19

**Authors:** Karolline S. May, Laura J. den Hartigh

**Affiliations:** 1Department of Medicine, Division of Metabolism, Endocrinology, and Nutrition, University of Washington, Seattle, WA 98109, USA; ksmay@u.washington.edu; 2UW Medicine Diabetes Institute, 750 Republican Street, Box 358062, Seattle, WA 98109, USA

**Keywords:** gut microbiota, obesity, host metabolism, acetate, propionate, butyrate, GLP-1, PYY, lipolysis, adipogenesis

## Abstract

Obesity and its complications—including type 2 diabetes, cardiovascular disease, and certain cancers—constitute a rising global epidemic that has imposed a substantial burden on health and healthcare systems over the years. It is becoming increasingly clear that there is a link between obesity and the gut microbiota. Gut dysbiosis, characterized as microbial imbalance, has been consistently associated with obesity in both humans and animal models, and can be reversed with weight loss. Emerging evidence has shown that microbial-derived metabolites such as short-chain fatty acids (SCFAs)—including acetate, propionate, and butyrate—provide benefits to the host by impacting organs beyond the gut, including adipose tissue. In this review, we summarize what is currently known regarding the specific mechanisms that link gut-microbial-derived SCFAs with adipose tissue metabolism, such as adipogenesis, lipolysis, and inflammation. In addition, we explore indirect mechanisms by which SCFAs can modulate adipose tissue metabolism, such as via perturbation of gut hormones, as well as signaling to the brain and the liver. Understanding how the modulation of gut microbial metabolites such as SCFAs can impact adipose tissue function could lead to novel therapeutic strategies for the prevention and treatment of obesity.

## 1. Introduction

The obesity epidemic has risen over the past several decades, and is associated with a plethora of metabolic disorders in both children and adults [1,2,3]. To date, the prevalence of overweight and obesity in the United States has passed 40%, which imposes a substantial health burden and excessive healthcare costs [4,5]. Epidemiological studies have shown that obesity-related complications—including cardiovascular disease (CVD), type 2 diabetes (T2D), liver disease, and certain types of cancer—subsequently contribute to escalating morbidity and mortality [6,7,8,9]. Hence, innovative approaches for the prevention and treatment of obesity are urgently needed [10].

During the onset of obesity, expansion of the adipose tissue is fostered by hyperplasia of pre-adipocytes and hypertrophy in adipocytes [11]—processes that are intended to increase the adipose tissue’s lipid-buffering capacity. However, concomitant increases in the production and secretion of pro-inflammatory markers—as well as ectopic fat storage in and around metabolic organs such as the liver, pancreas, and skeletal muscle—tend to promote metabolic dysfunction [12,13,14]. Taken together, functional obesity-related disturbances not only in the adipose tissue, but also in other metabolic organs, may cause disturbances in glucose metabolism, insulin signaling, and cardiometabolic disease [13,15].

In addition to the notion that excess caloric intake coupled with reduced energy expenditure drives obesity, compelling evidence suggests that an imbalance of the gut microbiota and associated host physiology can also modulate obesity in animal models [16,17,18,19] and humans [20,21], potentially by altering host energy metabolism. Gut microbiota composition is altered in people and mice with obesity, and is also altered by weight loss [22,23,24,25,26]. Germ-free mice are protected from diet-induced obesity and comorbidities such as insulin resistance [27,28], and the obesity phenotype can be conferred by transplantation of fecal contents from obese mice into lean, germ-free mice [16], suggesting that the “obese” microbiome is sufficient to cause obesity. The microbiota from obese individuals in particular has been generally characterized by a lower bacterial phylogenetic diversity [29]. In line with this, the loss of beneficial microbes and excessive growth of harmful microorganisms—which together contribute to reduced microbial diversity—could impact obesity and adipose tissue metabolism, in part by modulating microbial-derived metabolites such as short-chain fatty acids (SCFAs) [30,31,32]. The host diet has a profound impact on gut microbial composition, altering both species diversity and abundance [33,34,35,36]. Thus, it stands to reason that because the symbiotic relationship between the gut microbiota and host metabolism is heavily influenced by dietary components, it is very likely that obesity-associated gut dysbiosis could similarly impact host metabolism. Additional potential mechanisms by which particular gut microbes could impact obesity include (1) alteration of intestinal permeability to endotoxins, thereby contributing to the persistent low-grade systemic inflammation that is commonly observed in obesity [37], and (2) perturbation of the production of gut hormones that influence the gut–brain–adipose axis to alter the intake, storage, and utilization of energy [38,39]. For the purposes of this review, we will focus our attention on mechanisms by which microbial SCFAs could impact obesity.

SCFAs are linear fatty acids with fewer than six carbons, of which acetate (C:2), propionate (C:3), and butyrate (C:4) constitute >95% of all intestinal SCFAs [40], with an approximate molar ratio of 60:20:20, respectively, in the cecum and colon [41]. These metabolites—end-products of bacterial fermentation of non-digestible dietary fibers—are mainly produced in the gut to varying degrees, depending on the type of carbohydrate-based substrates available, along with the extent of their degradation and the specific bacterial strains that are present [40,42,43,44]. Acetate—the most abundant systemic SCFA—and propionate are predominantly produced by various genera in the phylum *Bacteroidetes*, whereas butyrate is the primary metabolite derived from some genera in the phylum *Firmicutes* [17,18,45]. Specific taxa that produce each of the three primary SCFAs are listed in Table 1.

While SCFAs exert well-documented local effects in the host gut, they are also absorbed and circulate systemically, where they exert putative effects in tissues and organs beyond the gut (Figure 1). Thus, SCFAs acting as signaling molecules that target metabolic organs may play important roles in metabolic diseases, including obesity [32]. Of note, a handful of studies have shown that acetate, propionate, and butyrate might alter adipose tissue function by enhancing adipogenesis, with a concomitant increase in the expression of genes related to adipogenic differentiation [46,47], attenuating intracellular lipolysis [48,49], and by reducing the release of pro-inflammatory cytokines and several other chemokines [50,51]. These effects will be discussed further in later sections.

Despite the considerable attention given to the therapeutic potential of SCFAs for obesity, the metabolic outcomes of altering SCFAs in longer term intervention studies remain unclear, as do their effects on systemic organs such as adipose tissue. As such, it is essential that we understand the mechanistic effects of SCFAs on adipocytes, due to the importance of adipose tissue as a metabolic regulator of whole-body energy homeostasis [52]. Thus, elicited interest in the underlying mechanisms by which SCFAs modulate adipocyte metabolism will be discussed in this review.

## 2. SCFA Receptors and Obesity

The major SCFA receptors known to date are the G-protein-coupled receptors GPR41 (also known as free fatty acid receptor 3 (FFAR3)) and GPR43 (FFAR2). GPR41 and GPR43 are related members of a family of GPCRs sharing approximately 42% homology, which are tandemly encoded at a single chromosomal locus in both humans and mice. GPR41 and GPR43 have previously been reviewed extensively [66]. Previously described as orphan GPCRs, they were identified as specific SCFA ligands nearly 20 years ago [67]. GPR41 and GPR43 are activated by micromolar concentrations of the SCFAs acetate, propionate, and butyrate. Differences in the affinity of different SFCAs for these receptors have been reported in various animal species. In humans, acetate exhibits high affinity for only GPR43, but in mice it can activate either GPR43 or GPR41 [68]. Propionate and butyrate have high affinity for both GPR41 and GPR43 [69]. Species differences have also been reported in bovines, raising the possibility that GPRs respond differently to SCFAs due to adaptation to particular nutritional regimens [70].

GPR41 and GPR43 have been shown to activate the G_i/o_ family proteins in ^35^S-GTPγS-binding assays using HEK294T cells transfected with either receptor, but GPR43 may also signal through G_q_ [67,71,72]. Stimulation of GPR43 by SCFAs then inhibits cAMP production, which increases intracellular Ca^2+^ levels, leading to subsequent activation of the mitogen-activated protein kinase (MAPK) pathway—particularly via the extracellular signal-regulated kinase (ERK) arm of the pathway [73]. ERK activation has been shown to be required for adipogenesis in cultured adipocytes due to its activation of the critical adipogenesis transcription factor CCAAAT/enhancer-binding protein beta (C/EBPβ) [74,75]. Thus, GPR41 and GPR43 could play important roles in adipocyte metabolism.

In humans, GPR41 has been detected in adipocytes, the spleen, the pancreas, lymph nodes, bone marrow, gut enterocytes, and enteroendocrine cells, as well as peripheral blood mononuclear cells (PBMCs) [67,71,76], while GPR43 is expressed largely in adipocytes, gut enterocytes, enteroendocrine cells, and PBMCs [67,71,72,77]. The expression patterns appear to differ considerably in mice, with GPR41 expressed primarily in the kidneys, colon, and spleen, and GPR43 expressed more ubiquitously in the adipose tissue, stomach, colon, spleen, and immune cells [78,79,80]. There is some inconsistency in the literature regarding whether GPR41 is expressed in adipose tissue [81]. Notably, GPR43 expression in white adipose tissue was confined to adipocytes, and not found in the stromal vascular fraction [78]. It is still unclear whether brown adipocytes express GPR43, with one group reporting no expression in BAT [82], and others reporting low expression levels in immortalized brown adipocyte cell lines, albeit at much lower levels than in white adipocytes [83,84].

Most studies examining the activation of GPR41 and GPR43 by SCFAs have been conducted using in vitro models and in mice. Such studies have revealed that GPR41 is primarily activated by SCFAs in gut-hormone-producing enteroendocrine cells [85]. GPR41 is also expressed by enteric neurons in the gut, which enables direct signaling between SCFAs and the brain [86,87], suggesting a direct energy-balance-regulatory mechanism by SCFAs. In addition, SCFAs that circulate systemically can readily activate GPR43 on white adipocytes, enabling a direct gut–adipose axis that can modulate energy metabolism. In the following sections we will discuss and evaluate previous studies that have examined the effects of SCFAs on adipose tissue metabolism.

### 2.1. Modulation of Obesity by SCFAs

There is a clear link between intestinal microbiota composition and obesity. Dysbiosis, or microbial imbalance in the host, has been associated with obesity in both humans and mice, and can be reversed by weight loss [22,24,28,85,88]. Germ-free mice that do not host gut microbes are protected from diet-induced obesity, and the obesity phenotype can be conferred by transplantation of fecal contents from obese mice into lean, germ-free mice [16,27,28], suggesting that the gut microbiome of such animals is sufficient to induce obesity.

Specific metabolites that are (or are not) produced by gut microbes in obesity—such as SCFAs—may modulate the obesity phenotype. While some studies have found that obesity is associated with elevated fecal SCFA levels in mice and humans [16,89,90], others have shown reduced fecal SCFA levels in diet-induced obese rodent models [17,91], highlighting a degree of uncertainty in the field regarding a clear role of SCFAs in obesity. However, obese mice have been shown to express higher levels of the SCFA receptors GPR41 and GPR43 in the colon and white adipose tissue than lean mice [78,92].

Dietary regimens that increase colonic SCFA levels have been shown to impart resistance to diet-induced obesity and adiposity [93], suggesting that colonic SCFAs signal either directly or indirectly to white adipose tissue (WAT) to modulate WAT metabolism. Mice that were fed a high-fat diet (HFD) containing 10% fermentable flaxseed fiber, which increased total SCFA levels, gained less body weight and body fat and exhibited improved glucose metabolism compared to HFD-fed mice, and exhibited similar energy metabolism to that of lean, chow-fed mice [93]. The authors observed a marked increase in colonic butyrate levels, which was associated with increased abundance of the genera *Lactobacillus*, *Akkermansia*, and *Bifidobacterium*—known lactate and butyrate producers [94,95,96]. Interestingly, these genera have also been implicated as producers of conjugated linoleic acid [97], which is also associated with a negative energy balance phenotype in diet-induced obese mice [98,99]. Thus, dietary strategies that enrich the gut microbiota with SCFA-producing genera could reduce body weight and improve comorbidities associated with obesity.

Another approach to increasing systemic SCFA levels is through direct dietary supplementation. Adding acetate, propionate, butyrate, or their mixture (3:1:1 ratio) (5% wt/wt each) to a HFD (60% kcal from fat) resulted in less body weight gain, with no differences in food intake in mice [17]. Epididymal WAT levels of GPR43 and GPR41 were increased by all SCFAs, while they tended to decrease in the colon [17], potentially in a compensatory manner. In addition, adiponectin levels increased in WAT in mice given SCFAs. SCFA supplementation increased expression of the genes encoding carnitine palmitoyltransferase 1-alpha (*CPT1α*), acetyl-CoA carboxylase (*ACC*), lipoprotein lipase (*Lpl*), and hormone-sensitive lipase (*Hsl*) mRNA expression in WAT, with no change in free fatty acid (FFA) plasma levels. SCFAs also increased peroxisome proliferator-activated receptor gamma coactivator-1 alpha (*PGC1α*) mRNA in epididymal WAT, suggesting a WAT browning effect of SCFAs given orally. Butyrate in particular has been shown to prevent diet-induced obesity and associated insulin resistance when given orally [100]. Thus, evidence to date suggests that microbial SCFAs could directly impact the obesity phenotype, potentially by modulating adipocyte metabolism via GPR41 or GPR43.

### 2.2. Perturbation of SCFA Receptors: The Impact on Obesity

In order to explore whether GPR41 or GPR43 are required for SCFA-mediated effects on energy balance, loss-of-function studies have been performed. While several GPR41- and GPR43-knockout models have been described to date, the majority of them focus on endpoints related to immunity, inflammation, and intestinal function in the gut and related cells [101,102,103,104,105,106,107,108,109,110,111]. In the following sections, we will summarize what is known about energy balance and adipocyte metabolism from animal models in which GPR41 or GPR43 have been perturbed genetically.

#### 2.2.1. GPR41

The first GPR41-knockout (KO) mice described weighed significantly less than their germ-free and WT counterparts, despite equivalent food intake [111]. This effect was only evident when microbes were present, implying that gut-microbe-derived metabolites that activate GPR41 promote body weight gain. Mice deficient in GPR41 were also found to exhibit reduced leptin and protein YY (PYY)—an enteroendocrine-cell-derived hormone that normally inhibits gut motility, increases intestinal transit rate, and reduces the harvest of energy (i.e., SCFAs) from the diet. Thus, the impact of reduced SCFA signaling to GPR41 may have been confounded by reduced efficiency of energy harvest in this model. Notably, GPR41-KO mice had elevated cecal and fecal acetate and propionate levels, suggesting reduced SCFA absorption from the gut. Thus, from this initial study, it is difficult to interpret the impact of global loss of GPR41 signaling on adipose metabolism.

Subsequent GPR41-KO mice were generated and reported to have normal growth, and body weights equivalent to wild-type control mice [112]. However, these GPR41-KO mice exhibited reduced body temperature and brown adipose tissue activation [112], suggesting that GPR41-mediated SCFA uptake plays an important role in adaptive thermogenesis. A third GPR41-KO mouse model displayed increased adiposity with reduced energy expenditure when fed either a low-fat or high-fat diet [113]. The reasons for the wide range of phenotypes observed thus far in GPR41-KO mice is not immediately clear, and warrants further investigation.

#### 2.2.2. GPR43

Mice deficient in GPR43 do not display an overt phenotype when fed a chow diet [53,113]. Acetate and G_i/o_ signaling have been shown to suppress lipolysis both in vitro and in vivo [53,114,115,116]—an effect that was blunted in GPR43-KO mice [113]. When fed an HFD, no differences in body weight or composition were noted until 26 weeks of age, at which point GPR43-KO mice were leaner and weighed less than their wild-type counterparts, likely due to increased energy expenditure [113]. This was despite increased food intake in mice deficient in GPR43, but was also coupled with increased fecal energy density, suggesting reduced feeding efficiency [113].

Conversely, in a separate study, GPR43-KO mice were found to have higher body weights than wild-type mice when fed either a chow or HFD, due largely to decreased energy expenditure and increased food intake [82]. Increased inflammatory profiles and worsened glucose and insulin tolerance were also observed in high-fat-fed GPR43-KO mice [82]. Importantly, the propensity for higher weight gain in GPR43-KO mice was abolished when the mice were reared under germ-free conditions, suggesting an important contribution by the gut microbiota [82].

Similarly, transgenic mice overexpressing GPR43 from adipocytes (driven by the AP2 promoter) displayed lower body weights and increased energy expenditure compared to non-transgenic mice, suggesting that adipocytes are an important target for SCFAs via GPR43 [82,117]. These mice were also resistant to HFD-induced obesity, due to increased energy expenditure with improved glucose and insulin tolerance [82]. Another model of GPR43 overexpression from adipocytes showed increased energy expenditure and fat oxidation—effects that were only observed in the presence of gut microbiota [117]. Coupled with the obesity-promoting phenotype of GPR43 deletion, studies to date suggest that GPR43 plays an anti-obesogenic role in adipocytes.

## 3. Direct Effects of Microbial-Derived SCFAs on Adipose Tissue

SCFAs, as bacterial metabolites produced in the gut that achieve relatively high levels in the circulation, have the capacity to directly signal to adipose tissue, where they have been shown in several studies to modulate adipose tissue metabolism. While SCFAs are produced by gut bacteria in an approximate 60:20:20 molar ratio for acetate, propionate, and butyrate, respectively, the levels observed in the circulation in mice and humans differ considerably. In humans, this molar ratio changes to 91:5:4, with the majority of systemic SCFAs deriving from acetate [41]. In contrast, the majority of butyrate produced by the gut microbiota is absorbed and utilized by colonocytes in the gut [40], with relatively lower abundance systemically. However, because SCFAs are produced in such large quantities in the intestine, what reaches the systemic circulation is still quite substantial, estimated at 100–400 mM levels per day, depending on the amount of fiber present in the diet [41,118]. In the following sections, the direct role of systemic SCFAs on adipocytes will be examined

### 3.1. Microbial-Derived SCFA Effects on Energy Metabolism and Glucose Homeostasis

SCFAs are key players in regulating host energy metabolism, providing ~10% of the total energy intake in humans [118]; however, previous studies have proposed that these compounds mitigate diet-induced body weight gain and adiposity [17,100,119,120,121,122]. In line with this, SCFAs can mediate energy balance by increasing energy expenditure and fat oxidation [17,100,119], stimulating intestinal gluconeogenesis [122,123] as well as gut-derived hormones [38,73,124], and can signal directly to WAT [125,126,127]. Furthermore, the sympathetic nervous system is also associated with control of energy balance and, therefore, impacts weight loss [128]. In this section, we will focus on the effects of SCFAs on energy metabolism, with particular attention given to glucose homeostasis, in human and rodent studies.

Evidence to date supports the notion that SCFAs may exert a beneficial rather than detrimental effect on host metabolism [32]. However, the available human in vivo data indicating the physiological effects of SCFAs are limited (see Table 2). Chambers et al. [129] showed that daily administration of 6.8 g of propionate acutely increased the release of gut hormones and reduced energy intake in healthy but overweight subjects. In parallel, a 24-week supplementation with 10 g of inulin-propionate ester significantly prevented body weight gain, followed by reduced intra-abdominal adipose tissue distribution in overweight adults [129]. Further analyses conducted by Canfora et al. [130] assessed the physiologically relevant colonic infusions of 200 mmol/L SCFA mixtures in overweight or obese normoglycemic men. In this randomized double-blind clinical investigation, acetate increased the secretion of the satiety-stimulating gut hormone PYY, fat oxidation, and resting energy expenditure, and decreased whole-body lipolysis, which translates into benefits for body weight control [130].

Conversely, a small pilot investigation examined the differential metabolic effects of daily oral administration of 4 g of butyrate in individuals who were lean or had metabolic syndrome. Whereas no differences were found in terms of energy expenditure between the groups, sodium butyrate markedly improved glucose metabolism in lean subjects, but not in subjects with metabolic syndrome, presumably due to altered SCFA handling and flux in insulin-resistant subjects [131]. By contrast, several studies in which mice were given oral acetate or butyrate reported reduced body weight and body fat due to increased energy expenditure and fat oxidation [100,132]. Propionate and butyrate improve glucose and insulin tolerance in rats when administered orally [122].

Extending these observations, in response to alterations in energy status, the adipose tissue may undergo metabolic imbalance, accompanied by dysregulation of adipocyte glucose homeostasis and insulin sensitivity [133]. SCFA treatment might affect those parameters either indirectly—via gut-derived hormones such as PYY and the incretin glucagon-like peptide 1 (GLP-1) (to be discussed in Section 4.1)—via alterations in hepatic glucose regulation [42,134,135], or by directly impacting adipose tissue glucose metabolism.

Randomized human trials have shown that acetate administration improves the triglyceride storage capacity in adipocytes as well as islet β-cell function. Infusion of 180 mmol/L acetate in men with overweight and obesity affected whole-body metabolism, as measured by the increase in fasting fat oxidation, PYY, and postprandial insulin and glucose levels [136]. In women with hyperinsulinemia, 60 mmol acetate administered rectally led to increased PYY and GLP-1 as a secondary outcome, with no changes in insulin and glucose concentrations [137]. These inconsistent findings reinforce the need for additional human intervention studies to understand the impact of SCFAs on glucose metabolism and insulin sensitivity, with a focus on adipose tissue.

Based on animal data, Yan et al. [54] showed that butyrate administered at various doses stimulates adipocyte differentiation and adiponectin expression in porcine stromal vascular cells by upregulating glucose uptake, with enhanced insulin sensitivity concomitant with downstream AMP-activated protein kinase (AMPK) pathway signaling. In line with this, Gao et al. [100] showed that dietary supplementation with butyrate reduces adiposity, counteracts insulin resistance, and leads to activation of AMPK in mice fed a high-fat diet, by promoting energy expenditure and mitochondrial function.

A few studies performed on cultured adipocytes suggest an effect of SCFAs on glucose metabolism. An initial study performed on ovine adipocytes revealed that acetate increased glucose uptake 1.5-fold, resulting in increased pentose phosphate pathway activity and increased fatty acid synthesis [138]. These enhanced effects on glucose uptake and fatty acid synthesis were confirmed in explanted adipose tissue from rats [139]. Studies in other cell types have shown that SCFAs can increase the expression of GLUT4—the major glucose transporter expressed in adipocytes [55]. In addition, SCFAs can activate AMPK [100,132,140], which initiates catabolic energy utilization pathways such as fat oxidation and glucose consumption. Thus, although studies to date are limited, it is plausible that SCFAs can directly modulate glucose metabolism in adipocytes.

In summary, the above studies provide clues for the experimental design and use of SCFAs in adiposity management and insulin sensitivity, which may depend on mechanistic biological approaches such as the mode of administration, targeted species-specific differences, and metabolic phenotypes [134,135,141,142,143].

### 3.2. SCFAs and Lipolysis

Lipolysis, a functional hallmark of adipocytes, is defined as a sequential breakdown of triacylglycerols (TAGs) into glycerol and non-esterified fatty acids (NEFAs) [148], in which several factors exert influence. These include β-adrenergic stimulation, insulin/glucagon signaling, activation of inhibitory/stimulatory G-protein-coupled receptors, and the synergy between these factors [125,126,127]. In this respect, evidence has pointed to an important role of SCFAs on the lipolytic pathways in white adipose tissue (see Table 3) [52,53,55,116,149]. In humans, acetate has consistently been associated with an antilipolytic phenotype [130,145,150]. In this section, we will present evidence that SCFAs modulate lipolysis in adipocytes in vitro.

Recent analyses of SCFAs in differentiated human multipotent adipose-derived stem cells demonstrated that mainly acetate, ranging from 1 μmol/L to 1 mmol/L, exerts an antilipolytic profile by attenuating hormone-sensitive lipase (HSL) at phosphor-Ser650, in a GPR-dependent manner [52]. Complimentary to these data, the presence of a supraphysiological concentration (10 mM) of acetate affects lipolysis by decreasing the phosphorylation of HSL563 and HSL562 in murine and human primary adipocytes, respectively, which also seems to involve GPR [151].

Other findings provide reasonable insight into the mechanisms of action of SCFAs on stimulated 3T3-L1 adipocytes. Aberdein et al. [116] showed that 4 mM acetate reduces the phosphorylation of HSL_(Ser563)_ in 3T3-L1 in conjunction with the rate of lipolysis, as mirrored by NEFA release. Furthermore, the incubation of differentiated 3T3-L1 adipocytes with either 0.1 μmol/L acetate or propionate in a dose-dependent manner inhibits lipolysis via GPR43 [78] Similarly, Ge et al. [53] showed that acetate and propionate within physiological concentrations (0.1–0.3 mmol/L) suppress lipolytic activity by up to 50% via the activation of GPR43 in vitro. As such, SCFAs bind and stimulate G-protein-coupled receptors, with acetate and propionate exhibiting high affinity for GPR43, followed by butyrate to a lesser extent. Such SCFA-mediated activation of GPR43 may decrease intracellular lipid overflow and regulate adipose tissue lipolysis [67,152].

Interestingly, an experiment carried out by Ohira et al. [55] revealed the dose-dependent lipolytic effect of butyrate (0.2–1 mmol/L) in co-cultured 3T3-L1 adipocytes and RAW264.7 macrophages, which can be explained by decreased lipase activity in adipocytes, as well as protein expression of adipose triglyceride lipase (ATGL), HSL, and phospho-HSL_Ser660_ [55]. Concurrently, the authors propose that butyrate appears to blunt lipolysis via GPR41, but not GPR43, in adipocytes. The interrelationship between adipocytes and distinct subpopulations of macrophages (inflammatory M1-type macrophages and anti-inflammatory M2-type macrophages) might explain the SCFA signaling/activation of one or another GPR, and its implications for metabolic activity [117].

Unlike the above studies, Rumberger et al. [149] reported in a long-term experiment that 20 mM propionate and 5 mM butyrate led to an increased rate of lipolysis in 3T3-L1 adipocytes. Nevertheless, while the precise mechanism of action of these end-products on adipocyte lipolysis remains unclear, the authors suggest that lower concentrations of SCFAs increase the lipolytic activity via alternative mechanisms other than the activation of GPR. In a sense, SCFA-mediated increases in lipolysis might be due to their histone deacetylase (HDAC)-inhibitory activity and changes in gene expression [149].

Whether these results are expected in cultured adipocyte precursor cells is not known. Further well-controlled studies should be performed in humans in order to better understand mechanistic pathways by which SCFAs influence lipolysis, with subsequent effects on adipose tissue metabolism.

### 3.3. SCFA Promote Adipogenesis and Browning of Adipose Tissue

SCFAs have been reported to influence the proliferation and differentiation of pre-adipocytes into mature adipocytes, as well as the browning of adipose tissue, which might represent therapeutic approaches for obesity. In this respect, Toscani et al. [153] described for the first time that 5 mM butyrate combined with insulin or dexamethasone may not only stimulate adipogenesis in 3T3-L1 cells, but also actively maintain the differentiation status in vitro. The stated effects were elicited by increasing the number of intracellular lipid droplets and the expression of the adipogenesis-associated genes lipoprotein lipase (LPL) and adipocyte fatty-acid-binding protein (aP2) [153], also known as FABP4.

Of note, varying concentrations of butyrate affect adipogenesis in porcine stromal vascular cells. Likely due to its HDAC-inhibitory activity, butyrate contributes to molecular changes on CCAAT- enhancer-binding proteins alpha/beta (C/EBPα/β) and peroxisome proliferator-activated receptor gamma (PPAR-γ), distinct transcriptional factors, and sterol regulatory element-binding protein 1c (SREBP-1c) [54]. To corroborate these findings, Li et al. [50] demonstrated that 0.1–3 mmol/L butyrate and propionate induce the differentiation of primary porcine adipocytes by ~20%, as well as greater expression of PPAR-γ and C/EBPα/β mRNA, in a time-sensitive manner. To some extent, the outcomes may be partially via butyrate’s HDAC-inhibitory activity, whereas those of propionate-stimulated cells are independent of their HDAC potential.

Another in vitro study has indicated the complimentary effects of butyrate, propionate, and acetate ranging between 0.8 and 6.4 mmol/L—non-cytotoxic concentrations—on the differentiation of 3T3-L1 adipocytes for 10 days, as confirmed by Oil Red O staining and enhanced gene expression of lipid metabolism markers, fatty acid synthase (FAS), fatty acid transporter protein 4 (FATP4), aP2, and LPL [154]. Hong et al. [78] provide insight into the distinct mechanisms of action whereby SCFAs contribute to adipogenesis. Incubation of 3T3-L1 cells with either 0.1 μmol/L acetate or propionate for 7 days upregulates PPAR-γ2 and GPR43 mRNA—an effect that was abrogated in the presence of GPR43 siRNA. Hence, these data suggest that GPR43 mediates SCFA-induced adipogenesis [78].

Remarkably, some analyses support the notion that SCFAs are not correlated with adipocyte differentiation. Alex et al. [155] revealed that 3T3-L1 and human Simpson–Golabi–Behmel syndrome (SGBS) adipocytes treated with high concentrations of propionate and butyrate (8 mM) induce neither PPAR-γ-mediated adipogenesis nor expression of cell differentiation markers. These findings are consistent with a more recent study in which physiological concentrations of propionate (0.01 mmol/L and 0.1 mmol/L) were unable to promote cell differentiation in chickens after an eight-day treatment, as confirmed by downregulated aP2 and PPAR-γ protein expression [156]. These seemingly conflicting data may be due to methodological differences, i.e., pre-adipocyte vs. mature adipocyte models, SCFA dosage, and other factors that may impair the direct comparison of different studies. See Table 4 for a list of studies related to SCFAs and adipogenesis.

SCFAs have also been implicated in the browning of WAT [17,51,84,146]. Brown adipose tissue (BAT) roughly comprises 2% of total stores in humans, and its presence is thought to counteract the development of obesity-associated complications [56]. In this context, a mechanistic study of obese individuals revealed that gut-microbial-derived acetate correlates with improved insulin sensitivity, energy expenditure, and weight loss assessed via increased gene expression of the browning markers PR domain containing 16 (PRDM16), uncoupling protein (UCP1), and iodothyrinine deiodinase 2 (DIO2) [146].

In another study, Hu et al. [84] aimed to evaluate whether SCFAs induce adipogenesis in immortalized brown adipocyte cells. Of note, the data show that 10 mM acetate leads to distinct morphological changes in brown adipocytes, an increase in adipogenesis and mitochondrial biogenesis, as well as the expression of PPAR-γ coactivator-1 alpha (PGC-1α), PPAR-γ, aP2, and UCP-1 through activation of GPR43 [84]. Accordingly, acetate (1 mM) alters beige adipogenesis-related genes in 3T3-L1 cells, with upregulation of cell-death-inducing DNA fragmentation factor-α-like effector A (CIDEA), transmembrane protein 26 (TMEM26), T-box transcription factor 1 (TBX1), FABP4, PPAR-γ, PRDM16, UCP-1, and DIO2 mRNA. Meanwhile, 6% acetate orally administered in drinking water for 16 weeks triggers fat oxidation and energy expenditure in obese KK-Ay mice, probably via GPR43 [51]. Lu et al. [17] confirmed the prior findings based on a chronic diet-induced obesity protocol in which male C57Bl/6J mice were supplemented with 5% acetate, propionate, and butyrate. Acetate promotes beige adipocyte differentiation and increases the expression of mitochondrial biogenesis-related genes by activating GPR43 in the adipose tissue [17]. Table 4 lists studies reporting the browning effects of SCFAs.

Thus, in order to better exploit the synergistic effects of SCFAs on adipogenesis and browning of adipose tissue, it is essential that we understand the mechanisms of action whereby SCFAs exert their effects beyond this organ. The combination of further in vivo and in vitro evidence is needed, which will guide interpretation of the findings to date.

### 3.4. SCFAs Modulate Adipose Tissue Inflammation

Obesity is associated with chronic low-grade inflammation, thereby contributing to adipose tissue dysfunction as well as the disturbance of host metabolic homeostasis [14,157,158]. An important hallmark of such chronic inflammation is increased infiltration and activation of macrophages, leukocytes, and other classes of immune cells into the adipose tissue, which is likely to drive insulin resistance and cardiovascular disease [11,13,56].

A few studies have contributed to what we know regarding the physiological relevance of SCFAs to adipose-tissue-related inflammation. As shown by Al-Lahham et al. [144], a 24-hour treatment with 3 mM propionate blunts basal cytokine production, including interleukin-10 (IL-10), interleukin-4 (IL-4), and tumor necrosis factor-α (TNF-α), as well as the chemokines macrophage inflammatory proteins-1α/β (MIP-1α/β) and C-C-chemokine ligand 5 (CCL5), in omental adipose tissue explants from overweight females. In addition, propionate favorably reduces the gene expression of macrophage-specific markers such as matrix metalloproteinase-9 (MMP9), CD163, and CD16A in adipose tissue explants [144].

In contrast, evidence suggests that SCFAs induce adipose tissue inflammation. Treatment with 10 mM acetate distinctly increases TNF-α expression in adipose tissue M2-type macrophages, but not M1-type macrophages in vitro [117]. Herein, under distinct stimuli, adipose tissue macrophages—M1, “classically activated macrophages” and M2, “alternatively activated macrophages”—are prone to inducing pro-inflammatory and anti-inflammatory secretory feedback, respectively [159,160,161].

With respect to the anti-inflammatory effects of SCFAs, butyrate protects adipose tissue of high-fat-fed mice against infiltration by leukocytes, attenuates interleukin-1β (IL-1β) and TNF-α expression, and restores the production of adiponectin [162]—an insulin-sensitizing adipokine. As such, these data indicate that SCFAs may counteract the infiltration of immune cells and overall inflammation-associated complications in obese adipose tissue [163].

Ohira et al. [55] described the potential of butyrate to mitigate monocyte chemoattractant protein 1 (MCP-1), interleukin-6 (IL-6), and TNF-α secretion in co-cultured 3T3-L1 adipocytes and RAW264.7 macrophages, in a dose-dependent manner. In a separate in vitro protocol, butyrate supplementation increased adiponectin expression in differentiated porcine stromal vascular cells, followed by the activation of downstream pathways, such as AMPK and the protein kinase B (AKT) [54].

Not only do SCFAs play an important role in white adipose tissue inflammation, they also have been shown to regulate the production of leptin—an important adipocyte-derived hormone directly involved in energy homeostasis. Dietary supplementation with a mixture of SCFAs or acetate increases leptin expression while reducing MCP-1 and IL-6 mRNA in the white adipose tissue of high-fat-fed mice [17]. Xiong et al. [125] found that 10 mM propionate increases leptin release in primary adipocyte cultures through the receptor GPR41. Consistently, ex vivo analysis has revealed that 3 mM propionate induces leptin in the order of 90%, probably via GPR41, but simultaneously mitigates the expression of resistin—a pro-inflammatory adipokine produced by macrophages, which bridges obesity to insulin resistance via GPR43 [127]. Another study conducted by Zaibi et al. [81] revealed that while acetate has an inhibitory effect upon leptin production, 3 mM propionate and 0.2 mM butyrate positively enhance the release of this adipokine in murine adipocytes mediated by GPR43 activation. Together, these findings support the possibility that either GPR41 or GPR43 modulates SCFA-stimulated leptin production in white adipose tissue.

## 4. Indirect Effects of SCFAs on Adipose Tissue

Due to reportedly high expression levels of GPR43—and in some cases GPR41—on adipocytes, there are likely direct effects of SCFAs on adipose tissue, as described in the previous section. However, it is also possible that indirect effects of SCFAs can also modulate adipose tissue metabolism. In this section, various potential indirect pathways by which gut-derived SCFAs could modulate adipose tissue function will be presented.

### 4.1. Gut–Brain Axis Regulated by SCFAs

GPR41 and GPR43 signaling contribute to gut homeostasis by impacting the gut–brain (reviewed in [164]), gut–liver and, potentially, gut–adipose axes [67]. GPR41 and GPR43 play emerging roles in the secretion of incretin hormones in the gut, such as PYY, GLP-1, and cholecystokinin (CCK), which all function to regulate food intake. Specifically, GPR41 and GPR43 expressed in enteroendocrine cells can modulate the SCFA-induced release of enteroendocrine hormones such as GLP-1, potentially due to their co-expression in colonic L-cells [121,165]. GPR41- and GPR43-KO mice exhibit reduced GPL-1 levels [165], consistent with their propensity for obesity and impaired insulin sensitivity.

While GLP-1 exerts known effects on glucose homeostasis, gastric emptying, and appetite regulation [166,167], its direct effects on adipocytes are not well described. White adipose tissue has been shown to express GLP-1 receptors in humans, and was inversely correlated with body weight and waist circumference in people with T2D and obesity [168]. However, a previous study showed that GLP-1 receptors were positively correlated with body weight and insulin resistance indices in subjects with morbid obesity [169]. The reasons for these discordant findings are not immediately clear, but may relate to the differences in subject characteristics. In vitro studies have shown that GLP-1 promotes adipogenesis in cultured 3T3-L1 adipocytes [170], suggesting a potential pathway by which SCFAs in the gut could stimulate GLP-1 release that can exert direct effects on adipocytes.

PYY is a gut hormone produced by enteroendocrine cells in the gut that functions to induce satiety, thereby reducing food intake [171]. Activation of GPR43 in the gut has also been shown to stimulate PYY secretion in humans and rodent models [103,172]. Mice fed an inulin-containing diet known to increase SCFA production by gut microbes [173,174] displayed increased PYY levels over controls—an effect that was duplicated in mice given SCFAs [85]. Notably, these effects were blunted in GPR43-KO mice, suggesting a synergistic effect of SCFA-mediated GPR43 activation and PYY secretion. Similarly to GLP-1, GPR41-KO mice display reduced PYY levels [111]. Subjects with obesity given propionate exhibited increased PYY and GLP-1 secretion, which was associated with significantly reduced adiposity and weight loss [129]. Collectively, these studies suggest that GPR41 and GPR43 are tightly linked with incretin signaling in the gut, which could complicate the interpretation of gain- and loss-of-function studies involving GPR41 and GPF43 on the regulation of adipose tissue metabolism.

In addition to their direct effects on gut hormones that are important for energy homeostasis, SCFAs may signal directly to the brain. Acetate and propionate have been found in the cerebrospinal fluid of healthy individuals at approximately 31 μM and 62 μM, respectively [134,175], with much lower concentrations of propionate and butyrate reported in human brain tissue (18.8 and 17.0 pg/mg of brain tissue, respectively) [176]. Moreover, both intravenous and colonic infusions with acetate increase the amount of acetate detected in the murine brain [120], suggesting that SCFAs can cross the blood–brain barrier. However, brain uptake of SCFAs appears to be more limited in humans and non-human primates [177,178]. GPR41 has been detected in the central nervous system, primarily localized to sympathetic ganglia and the vagal and dorsal roots [128,179]. Thus, while unlikely to play a prominent role in brain-mediated energy balance, SCFAs could exert direct effects on the brain.

### 4.2. Gut–Liver SCFA Axis That Impacts Adipose Tissue

As an organ directly downstream of blood flow leaving the gut via the portal vein, the liver is one of the first organs that may be impacted by gut metabolites such as SCFAs. This organ is also intimately involved with whole-body energy homeostasis, playing key roles in glucose and lipid regulation. The liver has been shown to be a key organ responsive to SCFAs. In cultured hepatocytes and in diet-induced obese mice, propionate and butyrate have been shown to decrease hepatic glucose production [180] and hepatic lipid accumulation [181,182], respectively.

Numerous studies attribute these effects to the ability of SCFAs to activate AMPK—an important regulator of glucose and lipid homeostasis in the liver. In murine models of hepatic steatosis, acetate and butyrate have been shown to activate AMPK-related signaling, with subsequently decreased hepatic steatosis due to increased fat oxidation [183,184].

Moreover, similar results have been observed in propionate- and butyrate-treated hepatocytes in culture [180,185]. Evidence in cultured hepatocytes suggests that these effects are mediated by GPR43 [180,185], but GPR43 expression has not yet been directly observed in the liver [186]. Thus, it is not clear whether SCFAs modulate hepatic glucose and lipid metabolism via a direct or indirect mechanism in vivo. It has also been shown in db/db mice that SCFAs can increase hepatic glucose, with subsequently increased glycogen synthesis, shown to require GPR43 in cultured HepG2 cells [185]. A recent study showed that GPR43 deficiency worsened insulin signaling only in the liver—not in skeletal muscle or white adipose tissue—an effect that was replicated using liver-specific delivery of GPR43 siRNA [187], suggesting a potential direct effect of acetate on the liver via GPR43. Moreover, propionate has been shown to suppress lipogenesis by decreasing fatty acid synthase in cultured hepatocytes [188].

Collectively, studies in mice and cultured hepatocytes suggest that SCFAs improve hepatic glucose and lipid metabolism, which could ultimately improve adipose tissue function in the context of metabolic dysfunction. Whether the effects of SCFAs on the liver are the result of direct or indirect effects remains to be elucidated.

## 5. Conclusions

Evidence to date supports the notion that microbial-derived metabolites provide beneficial rather than detrimental effects on obesity and its complications. These observed effects may be dependent on the nuanced experimental designs utilized in studies to date, including the method of SCFA administration, as well as targeted species-specific and metabolic phenotypes. SCFAs absorbed by the gut signal distinct metabolic responses to distant organs, including adipose tissue—an important regulator of whole-body energy homeostasis. Nevertheless, the metabolic phenotypes of altering SCFAs in long-term intervention protocols remain unclear. Further well-controlled studies should be carried out on genetically manipulated mice and in humans in order to better exploit the biological pathways by which SCFAs impact adipose tissue.

## Figures and Tables

**Figure 1 nutrients-13-03666-f001:**
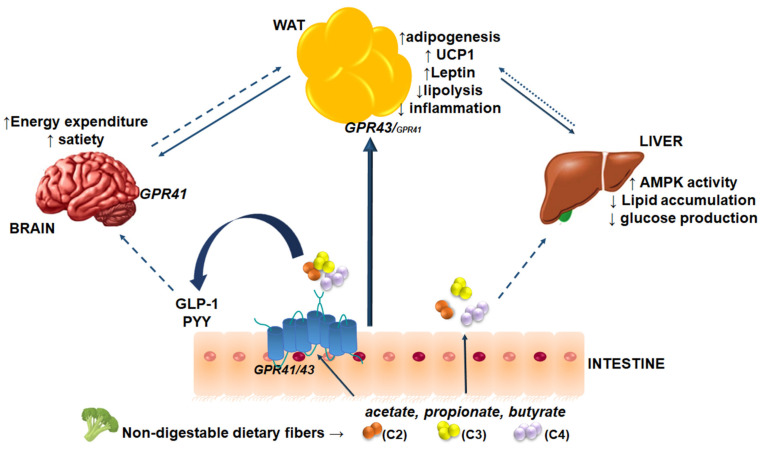
SCFA-mediated effects on host metabolism. Non-digestible dietary fibers consumed by the host are fermented by microbiota in the gut, resulting in the production of the SCFAs acetate (C2), propionate (C3), and butyrate (C4). These SCFAs are absorbed by diffusion or via GPR41 and GPR43 in the intestine. Systemic SCFAs can directly signal to adipose tissue to modulate adipogenesis, lipolysis, inflammation, browning, and adipokine synthesis. SCFAs can indirectly impact adipose tissue via effects on gut hormones, the brain, and the liver. In addition to directly targeting the brain via GPR41, SCFAs can also modulate the secretion of gut hormones such as GLP-1 and PYY, which target the brain to regulate food intake. SCFAs may also target the liver to reduce hepatic glucose production and lipid accumulation, likely through the activation of AMPK. Solid lines indicate direct effects of SCFAs via adipose tissue; dashed lines indicate indirect effects of SCFAs; dotted lines indicate hypothesized effects of SCFAs.

**Table 1 nutrients-13-03666-t001:** Taxa that produce SCFAs.

SCFA	Taxa	References	SCFA	Taxa	References
Acetate (C:2)	*Bifidobacterium* spp. *Lactobacillus* spp. *Ruminococcus* spp. *Prevotella* spp. *Streptococcus* spp. *Bacteroides* spp. *Akkermancia muciniphila* *Escherichia coli*	[53,54,55]	Butyrate (C:4)	*Bifidobacterium* spp. *Eubacterium* spp. *Coprococcus* spp. *Roseburia* spp. *Bacteroides* spp. *Anaerostipes* spp. *Clostridium leptum* *Clostridium butyricum* *Butyrivibrio fibrisolvens* *Butyricicoccus pullicaecorum* *Faecalibacterium prausnitzii*	[53,56,57,58,59,60,61,62,63]
Propionate (C:3)	*Bacteroides* spp. *Dialister* spp. *Veillonella* spp. *Salmonella* spp. *Lachnospiraceae* spp. *Megasphaera elsdenii* *Coprococcus cacus* *Roseburia inulinivorans* *Ruminococcus obeum* *Phascolarctobacterium succinatutens*	[54,56,64,65]			

Abbreviations—SCFAs: short-chain fatty acids; spp.: multiple species.

**Table 2 nutrients-13-03666-t002:** SCFAs modulate adipocyte metabolism in humans.

Experimental Design	Cell Type/Subject	Outcome	Study
Acetate given rectally (60 mM) or intravenously (20 mM), once weekly, for 4 weeks.	Hyperinsulinemic women (*n* = 6)	↑ PYY, GLP-1 (rectal) ↓ TNF-α ↔ Adiponectin	Freeland et al. 2010 [137]
WAT explants treated with 1–10 mM propionate for 24 h.	OAT and SAT explants from overweight women (*n* = 28)	↑ Leptin mRNA ↓ Resistin mRNA	Al-Lahham et al. 2010 [127]
WAT explants treated with 3 mM propionate for 24 h.	OAT explants from overweight women (*n* = 5) and THP-1 cells	↑ LPL, GLUT4, SREBP-1c mRNA ↓ MMP-9, CD163, CD16A, CD31 mRNA Propionate blunts IL-4, IL-10, TNF-α, MIP-1α, MIP-β, CCL5.	Al-Lahham et al. 2012 [144]
Intravenous acetate infusion (140 mM) for 90 min.	Men and women with (*n* = 3/6) or without (*n* = 4/5) hyperinsulinemia	↑ serum acetate ↓ FFA	Fernandes et al. 2012 [145]
10 g oral inulin-proionate ester (acute study). 10 g/day inulin-propionate ester for 24 weeks (chronic study).	Healthy men and women Acute effects (*n* = 15/5) Chronic effects (*n* = 60)	↓ Energy intake, intra-abdominal adipose tissue distribution, body weight, intrahepatic cholesterol ↑ PYY, GLP-1	Chambers et al. 2015 [129]
Colonic infusion with 200 mM SCFA mixtures for 4 days.	Normoglycemic men (*n* = 12)	↑ Energy expenditure, fat oxidation, PYY ↓ lipolysis	Canfora et al. 2017 [130]
Colonic infusion with 180 mM acetate infusions for 3 days.	Overweight/obese men (*n* = 6)	↑ Fasting fat oxidation, PYY, postprandial glucose and insulin	van der Beek et al. 2016 [136]
1 μmol/L–1 mM single SCFA (acetate, propionate, butyrate) or SCFA mixtures for 6 h.	hMADS	↓ Lipolysis, pHSL_(SER650)_ ↑ GPR41 and GPR43 mRNA and protein	Jocken et al. 2017 [52]
Plasma acetate measurement mRNA expression from WAT Fecal microbiota analyses	SAT and VAT explants from morbidly obese men and women (*n* = 6/28)	↑ Plasma acetate, browning markers ↓ GLUT-4 mRNA	Moreno-Navarrete et al. 2018 [146]
Oral delivery of butyrate (4 g/day) for 4 weeks	Healthy (*n* = 9) and metabolic syndrome men (*n* = 10)	↑ Hepatic insulin sensitivity in lean subjects No effect on energy expenditure and plasma/fecal butyrate concentration	Bouter et al. 2018 [131]
WAT explants treated with 3 mM propionate for 24 h	SAT explants from women (mean BMI = 28) (*n* = 10)	↓ TNF-α, IL-10, CD163, MMP-9 ↑ RANTES↑ LPL, GLUT-4, SREBP-1c	Al-Lahham et al. 2019 [147]

Relevant studies were identified using PubMed query searches for “short-chain fatty acids” OR “acetate” OR “propionate” OR “butyrate” AND “human” AND “adipose tissue” OR “adipocyte” AND “obesity”. Eleven studies examining the effects of SCFAs on adipocyte metabolism were thus identified. Abbreviations—SCFAs: short-chain fatty acids; PYY: peptide YY; GLP-1: glucagon-like peptide 1; TNF-α: tumor necrosis factor-α; LPL: lipoprotein lipase; SREBP-1c: sterol regulatory element-binding protein; hMADS: human multipotent adipose-tissue-derived stem cells; HSL; hormone-sensitive lipase; SAT: subcutaneous adipose tissue; VAT: visceral adipose tissue; OAT: omental adipose tissue; MIP-1α/β: macrophage inflammatory proteins-1α/β; CCL5: C-C-chemokine ligand 5; FFA: free fatty acid; GLUT4: glucose transporter type 4; ↑: increased; ↓: decreased.

**Table 3 nutrients-13-03666-t003:** SCFAs modulate adipocyte lipolysis: in vitro studies.

Experimental Design	Cell Type	Outcome	Study
1 μmol/L–1 mmol/L single SCFA (acetate, propionate, butyrate) or SCFA mixtures for 6 h.	hMADS	Acetate and SCFA mixtures high in acetate decreased lipolysis in basal human adipocytes.	Jocken et al., 2017 [52]
0.1–0.3 mmol/L acetate and propionate for up to 4 h.	3T3-L1, primary mouse adipocytes	Acetate and propionate inhibited lipolytic activity by approximately 50% in both cell types.	Ge et al., 2008 [53]
0.1–1 mmol/L butyrate for 24 h.	3T3-L1, RAW264.7 murine macrophages	Butyrate reduced lipolysis in co-cultured adipocytes in a dose-dependent manner.	Ohira et al., 2013 [55]
20 mmol/L acetate or 5 mmol/L butyrate for up to 4 h.	3T3-L1	Both acetate and butyrate increased the rate of lipolysis in a dose-dependent manner.	Rumberger et al., 2014 [149]
4 mM acetate for up to 2 h.	3T3-L1	Acetate decreased ISO-mediated lipolysis in adipocytes, mirrored by NEFA release in a time-dependent manner.	Aberdein et al., 2014 [116]
3–10 mM acetate, propionate and butyrate for 10 min.	Primary human and rat adipocytes	Propionate and butyrate inhibited lipolysis in human adipocytes, whereas acetate exerted an antilipolytic effect in primary rat adipocytes.	Heimann et al., 2015 [151]
0.01–0.1 μmol/L acetate and propionate for 7 days.	3T3-L1	Acetate and propionate blunted ISO-induced lipolysis in adipocytes, in a dose-dependent manner.	Hong et al. 2005 [78]

Relevant studies were identified using PubMed query searches for “short-chain fatty acids” OR “acetate” OR “propionate” OR “butyrate” AND “adipocyte” OR “3T3-L1” AND “lipolysis”. Seven studies examining the effects of SCFAs on adipocyte lipolysis were thus identified. Abbreviations—SCFAs: short-chain fatty acids; hMADS: human multipotent adipose tissue-derived stem cells; NEFAs: non-esterified fatty acids; ISO: isoproterenol. Fully differentiated 3T3-L1 and/or primary cells were used in the above-listed studies.

**Table 4 nutrients-13-03666-t004:** SCFAs promote adipogenesis and browning of adipose tissue.

Experimental Design	Cell or Tissue Type	Outcome	Study
5 mM butyrate for up to 7 days.	3T3-L1	↑ Adipogenesis ↑ Lipid droplets ↑ LPL, aP2 mRNA expression	Toscani et al. 1990 [153]
0.1–1.5 mM butyrate for up to 9 days.	Porcine stromal vascular	↑ adipogenesis ↑ C/EBPα/β, PPAR-γ, SREBP-1c mRNA	Yan et al. 2015 [54]
0.1–3 mM butyrate and propionate on day 10 of differentiation.	Porcine stromal vascular	↑ Adipogenesis ↑ C/EBPα/β, PPAR-γ mRNA	Li et al. 2014 [50]
6.4 mM acetate, 3.2 mM propionate, 0.8 mM butyrate for 10 days.	3T3-L1	↑ Adipogenesis ↑ FAS, FATP4, aP2, LPL mRNA	Yu et al. 2018 [154]
0.1 μmol/L acetate for 7 days.	3T3-L1	↑ Adipogenesis ↑ PPAR-γ2 mRNA	Hong et al. 2005 [78]
8 mM propionate and butyrate ~11 days.	SGBS	No effects on adipogenesis or adipogenic markers	Alex et al. 2013 [155]
0.01–0.1 mM propionate for 8 days.	Broiler adipocytes	No effects on adipogenesis ↓ aP2, PPAR-γ mRNA	Li et al. 2021 [156]
Plasma acetate measurement	SAT and VAT adipose tissue	↑ Browning adipogenesis ↑ PRDM 16, UCP1, DIO2 mRNA	Moreno-Navarrete et al. 2018 [146]
10 mM acetate for up to 7 days.	IM-BAT	↑ Browning adipogenesis ↑ Mitochondrial biogenesis ↑ PGC-1α, PPAR-γ, aP2, UCP-1 mRNA	Hu et al. 2016 [84]
1 mM acetate 0.6% acetate in drinking water for 16 weeks.	3T3-L1 KK-Ay mice	↑ Browning adipogenesis ↑ CIDEA, TMEM26, TBX1, FABP4, PPAR-γ, PRDM16, UCP-1, DIO2 mRNA	Hanatani et al. 2016 [51]
5% wt wt^−1^ acetate, propionate, butyrate or mixture SCFA supplemented (3:1:1 ratio) in high-fat diet for 16 weeks.	C57Bl/6J mice	↑ Browning adipogenesis ↑ PGC-1α, NRF-1, Tfam, β-F1 ATPase, COX IV, cyt-c	Lu et al. 2016 [17]

Relevant studies were identified using PubMed query searches for “short-chain fatty acids” OR “acetate” OR “propionate” OR “butyrate” AND “adipocyte” OR “3T3-L1” AND “adipogenesis” OR “browning”. Eleven studies examining the effects of SCFAs on adipocyte adipogenesis and/or browning were thus identified. Abbreviations—aP2: adipocyte fatty-acid-binding protein; β-F1: b subunit of the mitochondrial H^+^- ATP synthase; C/EBPα/β: CCAAT- enhancer-binding proteins; CIDEA: cell-death-inducing DNA fragmentation factor-α-like effector A; COX IV: cytochrome c oxidase IV; cyt-c: cytochrome c somatic; SGBS: human Simpson–Golabi–Behmel syndrome cells; DIO2: iodothyrinine deiodinase 2; FAS: fatty acid synthase; FATP4: fatty acid transporter protein 4; LPL: lipoprotein lipase; NRF: nuclear respiratory factor 1; PPAR-γ: peroxisome proliferator-activated receptor gamma; PRDM16: PR domain containing 16; SREBP-1c: sterol regulatory element-binding protein 1c; TBX1: T-box transcription factor 1; Tfam: mitochondrial transcription factor A; TMEM26: transmembrane protein 26; UCP1: uncoupling protein; SAT: subcutaneous adipose tissue; VAT: visceral adipose tissue; IM-BAT: immortalized brown adipocyte cells. ↑: increased; ↓: decreased.

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
