# Peer review of "Modulation of Adipocyte Metabolism by Microbial Short-Chain Fatty Acids"

_nutrients, 2021, doi:10.3390/nu13103666_

Round 1
Reviewer 1 Report
Reviewers
General comment- This is a clearly presented and well-written review from Dr. Hartigh’s group. In this manuscript, the authors reviewed the role of gut microbial metabolites especially short chain fatty acids (SCFA) on adipose tissue metabolism (adipogenesis, lipolysis and inflammation). It has been well reported in literature that gut dysbiosis has been associated with obesity in mice and human settings; however, whether gut derived SCFAs have beneficial effects to host beyond the gut is not much known.
Obesity is associated multiple complications such as Type 2 diabetes, cardiovascular problem, NAFLDs and certain cancers. Obesity is prevalent in 40% of the US population; however, the molecular mechanisms that drive this transition from being obese to particular disease complication is still unclear. Gut microbiome in obese murine models can cause obesity. In this article, authors have done exhaustive literature review how gut-derived SCFA can contribute to obesity and highlighted the role of gut microbiome on adipose tissue metabolism.
This review is very interesting, but I have following comments.
Authors have performed extensive literature search showing each relevant finding nicely explained and clearly demonstrated in tabulated forms about SCFA role in adipogenesis, lipolysis etc. and one figure.
It will be great if you can show all the relevant human work on gut microbiome (SCFA) role in adipose tissue metabolism (cell line/obese patients/diabetes/NAFLD) in either table or figure format.
Author Response
We thank Reviewer 1 for these kind words.
Reviewer 1 raises a great point about the importance of including a table highlighting the relevant human work related to SCFA and adipose tissue metabolism. We now have included a new Table (Table 2) listing human studies and interventions related to SCFA and adipocyte metabolism. Thank you for this suggestion- it will add translational appeal to our manuscript.
Reviewer 2 Report
The author wrote a review on the topic of “Modulation of adipocyte metabolism by microbial short-chain fatty acids”. It is well written as a whole, but the explanation of the gut microbiota is very insufficient. Since all strains included in the phylum level are not associated with SCFA-producing bacteria, it is necessary to write in detail, such as suggesting a representative genus rather than the phylum level. This can also provide information by organizing the contents of gut microbiota and SCFA into a table. Above all thing, SCFA-medicated effects on host mechanism including gut microbiota should be presented, because the author describes microbial-derived SCFA in the title and text.
Author Response
Reviewer 2 raises a great point, that our introduction to the gut microbiota and associated SCFA production could use improvement. We have now carefully edited the Introduction Section to include more relevant background information related to gut microbiota in obesity, as well as the effect of dietary components on microbiota composition (pages 2-3). We also included an introduction to how the gut microbial composition contributes to SCFA production, as well as new Table 1, which lists many genera previously reported to produce acetate, propionate, and butyrate.